_PLOS_ ONE

# Acute kidney injury during an ultra-distance race

Romain Jouffroy[1,2,3,4,5]*, Xavier Lebreton[6], Nicolas Mansencal[7,8], Dany Anglicheau[6]

**1** Intensive Care Unit, Anesthesiology, SAMU, Necker Enfants Malades Hospital, Assistance Publique - Hôpitaux de Paris, Paris, France, **2** Department of Anesthesia, Clinical Epidemiology and Biostatistics, Michael DeGroote School of Medicine, Faculty of Health Sciences, McMaster University, Hamilton, Ontario, Canada, **3** Population Health Research Institute; David Braley Cardiac, Vascular and Stroke Research Institute, Perioperative Medicine and Surgical Research Unit, Hamilton, Ontario, Canada, **4** Laboratory of Cellular and Molecular Mechanisms of Hematological Disorders and Therapeutical Implications - INSERM U1163 - ERL8254 – CEREMAST - Foundation IMAGINE, Paris, France, **5** IRMES - Institute for Research in Medicine and Epidemiology of Sport, INSEP, Paris, France, **6** Necker-Enfants Malades Institute, French National Institute of Health and Medical Research U1151, Paris Descartes, Sorbonne Paris Cité University, Department of Nephrology and Kidney Transplantation, Necker Hospital, Assistance Publique-Hôpitaux de Paris, Paris, France, **7** Department of Cardiology, Ambroise Paré Hospital, Assistance Publique-Hôpitaux de Paris (AP-HP), Centre de Référence des Cardiomyopathies et des Troubles du Rythme Cardiaque Héréditaires ou Rares, Université de Versailles-Saint Quentin (UVSQ), Boulogne, France, **8** INSERM U-1018, CESP, Team 5 (EpReC, Renal and Cardiovascular Epidemiology), UVSQ, Villejuif, France

* romain.jouffroy@gmail.com

## Abstract

### Purpose

Previous studies have noted consequences of ultra-distance trail running on health, but few studies are available regarding the temporal variations of renal biomarker injury during the running. The aim of this study was to assess the of kidney function parameters temporal variation during and on short-term after an ultra-distance race.

### Methods

We performed an observational study with 47 subjects participating in an ultra-distance race (80 km). Urine (47 subjects) and blood (21 subjects) samples were serially collected before (baseline—km 0), during (21 and 53 km), on arrival (80 km), and 9 days after the race (d9).

### Results

Mean serum creatinine increased during the race from 90±14 µmol/L (km0) to 136±32 µmol/L (km 80—p<0.0001) corresponding to a 52% increase. Mean creatininuria progressively increased from 4.7±4.5 mmol/L (km 0) to 22.8±12.0 mmol/L (km 80) (p<0.0001). Both urinary biomarkers (Neutrophil Gelatinase Associated Lipocalin and Kidney Injury Molecule-1) of acute kidney injury (AKI) progressively increased during the race (p<0.05 vs baseline). However, after adjustment to urine dilution by urine creatinine, no significant changes remained (p>0.05). On day 9, no significant difference remains in blood and urine biomarkers compared to their respective baseline levels.

**Data Availability Statement:** All relevant data are within the manuscript.

**Funding:** None author received funding for this work. No funding was allowed for the study design, data collection and analysis, decision to publish, or preparation of the manuscript.

**Competing interests:** The authors have declared that no competing interests exist.

## Conclusions

During an ultra-distance race, despite an acute and transient increase in the serum creatinine levels, urinary biomarkers of AKI displayed only limited changes with a complete regression on day 9. These results suggest the absence of the short-term impact of an ultra-distance race kidney function.

## Introduction

Practice of high-endurance sports, especially running, has increased in the last two decades.

In 2016, in the United States, more than 64 million runners participated in a marathon [1]. The beneficial effects of regular physical activity, especially on cardiovascular health, are well-established [2]. Physical activity also prevents lifestyle-related metabolic diseases [3] and results in colorectal cancers occurrence [4]. However, very little data exist on the health-related effects of ultra-endurance running, mainly focused on the cardiovascular field [5–8]. Contrary to marathons, trail running is a running race taking place in the wild (forest, plain, mountain) on a long-distance, on a generally marked course with positive and negative denivelation. Most of the time, this is a race in semi or total self-sufficiency requiring the wear a backpack with his food and drink supplies. "Ultra" defines a distance greater than 42 kilometers.

For years, renal function impairment has been defined in two distinct entities, i.e., acute and chronic kidney diseases. This dichotomic classification suggested that acute and chronic renal injuries were, ostensibly, not interconnected. However, during the two past decades, evidence from epidemiological and experimental studies has emerged suggesting that acute kidney injury (AKI) may trigger chronic kidney disease [9]. Independently of its cause, AKI is recognized as an important risk factor for persistent renal function impairment, incident chronic kidney disease, and accelerated progression to end-stage renal disease [9–14]. The ultimate consequence of the acute injury depends on the balance between the results of repair and regenerative pathways. The last one includes cell-cycle arrest, apoptosis, dedifferentiation, inflammatory infiltration, epigenetic factors, and profibrotic changes [9]. Maladaptive repair and disordered regeneration have both been suspected to be contributing mechanisms that may link acute injury to chronic lesions following AKI. Although a number of pathological conditions are well-known risk factors for AKI, more subtle and repeated insults might also trigger acute renal injury, which may worsen long-term renal function [15].

To date, the potential short-term deleterious effect of ultra-distance running on renal injury remains unclear. Previous studies observed an impairment of kidney function by comparing kidney function biomarkers values before and after the race [16–20]. The real temporal variations of the AKI occurrence during an ultra-distance race and the short-term impact of long-distance running are not known.

To help close this knowledge gap, we took advantage of an ultra-distance race in healthy subjects by combining serial blood and urine samples before, during and after an ultra-distance race to assess the temporal variations of acute renal injury during the race and short-term impact.

## Methods

### Participants

We performed an observational study involving 47 participants of the 80-km Ecotrail of Paris Ile-de-France race on 2014, March 29[th]. These 47 volunteers were electronically recruited using an announcement on the race's website (www.traildeparis.com).

Inclusion criteria were adult (age >18 years), male gender, and completion of an ultra-distance race (distance >50 km) during the last 12 months. Subjects with a history of chronic medical disease, e.g. high blood pressure or asthma, were not included.

During and after the race, the participants had free access to food and water, i.e. "ad libitum", limited to the refreshment points allowed by the race organisation. Participants exclusively drink water (no sport drinks) during the race.

Sports exercise, including running, was permitted between the end of the race and nine days after the end of the race (day 9).

The race's organizing committee, the French committee on public safety Paris Ile-de-France IV approved the protocol (Reference: 2014/07) as did the National Heart Agency (Number EudraCT: 2014-A00205-42) on 14 March 2014. All participants gave their written informed consent for participation before the start of the race.

## Study design and sample processing

Blood and urine samples were collected before the race (<24 hours corresponding to baseline or km 0), during the race at intermediate checkpoints (km 21 and km 53), at arrival (km 80) and on day 9. A nurse collected blood samples from 21 of the 47 subjects and urine samples from all subjects in order to assess serum electrolytes and biomarkers of renal function. The sampling performed during the race required a 5-minute stop at each checkpoint.

Urine and blood specimens were collected and immediately stored on ice and sent to Necker hospital (Paris, France). The urine samples were centrifuged at 1,000 g for 10 minutes. The supernatant was collected after centrifugation and stored with protease inhibitors at -80˚C. Frozen aliquots of urine supernatants were tested by ELISA for Kidney Injury Molecule-1 (KIM1, Quantikine ELISA #DKM100, R&D Systems) and for neutrophil gelatinase associated lipocalin (NGAL, NGAL ELISA kit, #036, BIOPORTO Diagnostics) according to the manufacturer's instructions. Measurement of urine creatinine was performed in the same sample using a Hitachi 917 analyzer (Roche Diagnostics). The results were normalized or not to the urinary creatinine level, and consequently, four biomarkers of AKI were analyzed:KIM1 (pg/mL), KIM1:creatinine ratio (KIM1:Cr, expressed in pg protein/mmoL urine creatinine), NGAL (pg/mL) and NGAL:creatinine ratio (NGAL:Cr, expressed in pg protein/mmoL urine creatinine).

The blood samples were immediately analyzed (analyzer Cobas—Roche Diagnostics) for plasma level of sodium, potassium, chloride, urea, creatinine, protein, C reactive protein (CRP), myoglobin, creatine kinase MB isoenzyme (CK-MB) and lactate.

## Statistical analysis

Continuous variables are presented as the mean ± standard deviation (SD). Categorical data are presented as absolute values and percentages. The distribution of each protein biomarker exhibited considerable positive skewness substantially reduced by use of a natural logarithm transformation.

We compared the levels of urinary protein biomarkers across the different time points using the Kruskal–Wallis test followed by Dunn's post-test. Additional analyses were performed using the Wilcoxon matched-pairs signed rank test to take into account the repeated measure design of the study. A p value <0.05 was considered to be statistically significant.

Statistical analyses were performed with GraphPad Prism (version 6.0f; GraphPad Software, San Diego, CA).

**Table 1. Characteristics of the studied population.** Data are expressed as mean ± standard deviation (SD) excepting time of race which is expressed as mean ± SD (minimal–maximal values).

| Parameter | Value |
|---|---|
| Age (years) | 43 ± 7 |
| Body weight (kg) | 74 ± 8 |
| Height (cm) | 176 ± 7 |
| Weekly covered distance (km) | 46 ± 18 |
| Weekly training duration (hours) | 5 ± 3 |
| Seniority in running practice (years) | 6 ± 4 |
| Number of trails per year | 5 ± 3 |
| Time of race (hours) | 11.2 ± 0.9 (9.1–12.4) |

## Results

The race took place on 2014, March 29[th] with a clear weather without rain with a temperature of +14˚C.

### Volunteer's parameters and performance

The demographic characteristics of the study population are presented in Table 1.

All 47 subjects reached the race finish line (no follow-up missing) and were examined at all checkpoints and on day 9.

The mean race completion time was 11.2 ± 0.9 hours (9.1–12.4) corresponding to an average race speed of 7.4±0.8 minutes/km. All subjects were trained with a week training mean time of 5 ± 3 hours corresponding to 46 ± 18 km per week.

### Blood laboratory data

Blood laboratory data are presented in Table 2.

Mean urea plasma levels increased during the race from 5.8 ± 1.6 mmol/L (km 0) to 9.2 ± 2.6 (p = 0.0005) (km 80), corresponding to a 59% increase from baseline. Mean creatinine plasma levels also increased during the race from 90 ± 14 μmol/L (km 0) (p<0.0001) to

**Table 2. Blood parameters in 47 runners at baseline, during the race (km 21 & 53), on arrival and on day 9.** Data are expressed as mean ± SD.

| Parameter | km 0 (baseline) | km 21 | km 53 | km 80 (arrival) | Day 9 |
|---|---|---|---|---|---|
| Na (mmol/L) | 143 ± 5 | 142 ± 2 | 145 ± 5 | 141 ± 7 | 141 ± 5 |
| K (mmol/L) | 4.8 ± 0.5 | 4.9 ± 0.6 | 4.7 ± 0.5 | 4.6 ± 0.5 | 4.8 ± 0.4 |
| Cl (mmol/L) | 104 ± 4 | 102 ± 3 | 101 ± 3 | 100 ± 5 | 101 ± 4 |
| Urea (mmol/L) | 5.8 ± 1.6 | 6.2 ± 1.6 | 8.3 ± 1.8 | 9.2 ± 2.6 | 5.9 ± 0.7 |
| Creatinine (μmol/L) | 90 ± 14 | 111 ± 22 | 136 ± 32 | 119 ± 33 | 90 ± 12 |
| eGFR (ml/min/m$^2$) | 89.1 ± 16.1 | 71.4 ± 15.8 | 57.7 ± 15.5 | 68.8 ± 21.4 | 88.3 ± 12.8 |
| Proteins (g/L) | 76 ± 10 | 78 ± 3 | 76 ± 6 | 72 ± 5 | 72 ± 3 |
| CRP (mg/L) | 1.1 ± 0.2 | 1.4 ± 0.4 | 2.5 ± 1.5 | 6.9 ± 4.3 | 2.9 ± 1.3 |
| Myoglobin (μg/L) | 34 ± 16 | 205 ± 148 | 1810 ± 2121 | 2761 ± 1484 | 2130 ± 1893 |
| CK-MB (ng/mL) | 13 ± 5 | 14 ± 6 | 32 ± 24 | 36 ± 34 | 96 ± 122 |
| Lactate (mmol/L) | 2.3 ± 0.8 | 3.0 ± 1.1 | 2.9 ± 1.2 | 3.3 ± 1.1 | 1.8 ± 0.7 |

Na = sodium, K = potassium, Cl = chloride, CRP = C reactive protein, CK-MB = creatine kinase MB isoenzyme.

136 ± 32 μmol/L (km 80) (p<0.0001). Serum creatinine peaked after 53 km, corresponding to a 52% increase from baseline.

Referring to the Kidney Disease Improving Global Outcome (KDIGO), Acute Kidney Injury (AKI) definition based on serum creatinine increase [16, 17], during the race (comparison with baseline), on km 21: 2 subjects (4%) met the stage 1 (2%) and none met the stages 2 or 3; on km 53: 5 (11%) subjects met the stage 1, 1 (2%) met the stage 2 and none met the stage 3; on km 80: 3 (6%) subjects met the stage 1 and none met the stages 2 or 3 and on day 9: no subject met the stage 1, 2 or 3.

The plasma level of myoglobin and CK-MB increased during the race without reaching significance. No other significant change of blood laboratory parameters was observed during the race.

## Urine laboratory data

Despite high interindividual variations, creatininuria progressively increased during the race from 4.7 ± 4.5 mmol/L (km 0) to 22.8 ± 12.0 mmol/L (km 80 p<0.0001) (Fig 1).

Longitudinal evolution of NGAL, NGAL:Cr, KIM1 and KIM1:Cr is summarized in Table 3 and Fig 2.

Urinary concentration of NGAL (Fig 2A) and KIM1 (Fig 2C) followed a similar dynamic pattern and significantly increased during the race.

During the race, NGAL values were, respectively, 15164 ± 19919 pg/mL (km 0), 39330 ± 34848 pg/ml (km 21), 75684 ± 65368 pg/ml (km 53, p<0.0001), and 38447 ± 41023 pg/ml (km 80, p<0.0001) (Table 3).

During the race, KIM1 values were 171.6±177.5 pg/ml (km 0, p<0.0001), 778.2 ± 1156 pg/ml (km 21, p<0.0001), 2204 ± 2222 pg/ml (km53, p<0.0001) and 1875 ± 2212pg/ml (km 80, p<0.0001).

Nine days after the race, NGAL and KIM1 urinary levels decreased but remained significantly higher than baseline levels (15164 ± 19919 pg/mL at baseline versus 24940 ± 28164 pg/mL at day 9, p = 0.007 for NGAL and 171.6 ± 177.5 pg/mL at baseline versus 672.7 ± 551.7 pg/mL at day 9, p<0.0001 for KIM1).

Taking into account the dilution status of the urine samples, the AKI urinary biomarkers displayed only limited variations over time. NGAL:Cr ratio actually decreased over time with similar urine levels at baseline and day 9 (Fig 2B).

KIM1:Cr ratio did not significantly increase at km-80 and returned to baseline levels at Day 9 (Fig 2D).

Considering the repeated measures design and that the samples are not independent, the Wilcoxon matched-pairs signed rank test confirmed the stability of the KIM1:Cr ratio between Km-0 and day 9 (p = 0.66) and even showed the decrease of the NGAL:Cr ratio (p = 0.04) between baseline and day 9 (Fig 3).

## Discussion

In this study, we observed a marked increase in the serum creatinine levels contrasting with limited change of urinary biomarkers of AKI during an ultra-distance race. Nine days after the end of the race, all kidney function biomarkers returned to their respective baseline levels. These results highlight that an ultra-distance race does not impact short-term kidney function.

In the last 10 years, ultra-trail has gained in popularity with a significant increase in races' length, but little is known regarding its health impact. These last ones were mainly explored in the cardiovascular field [5–7]. To date, ultra-trail is widely practiced by amateur runners. A number of deleterious effects described including acute myocardial infarction, acute cardiac

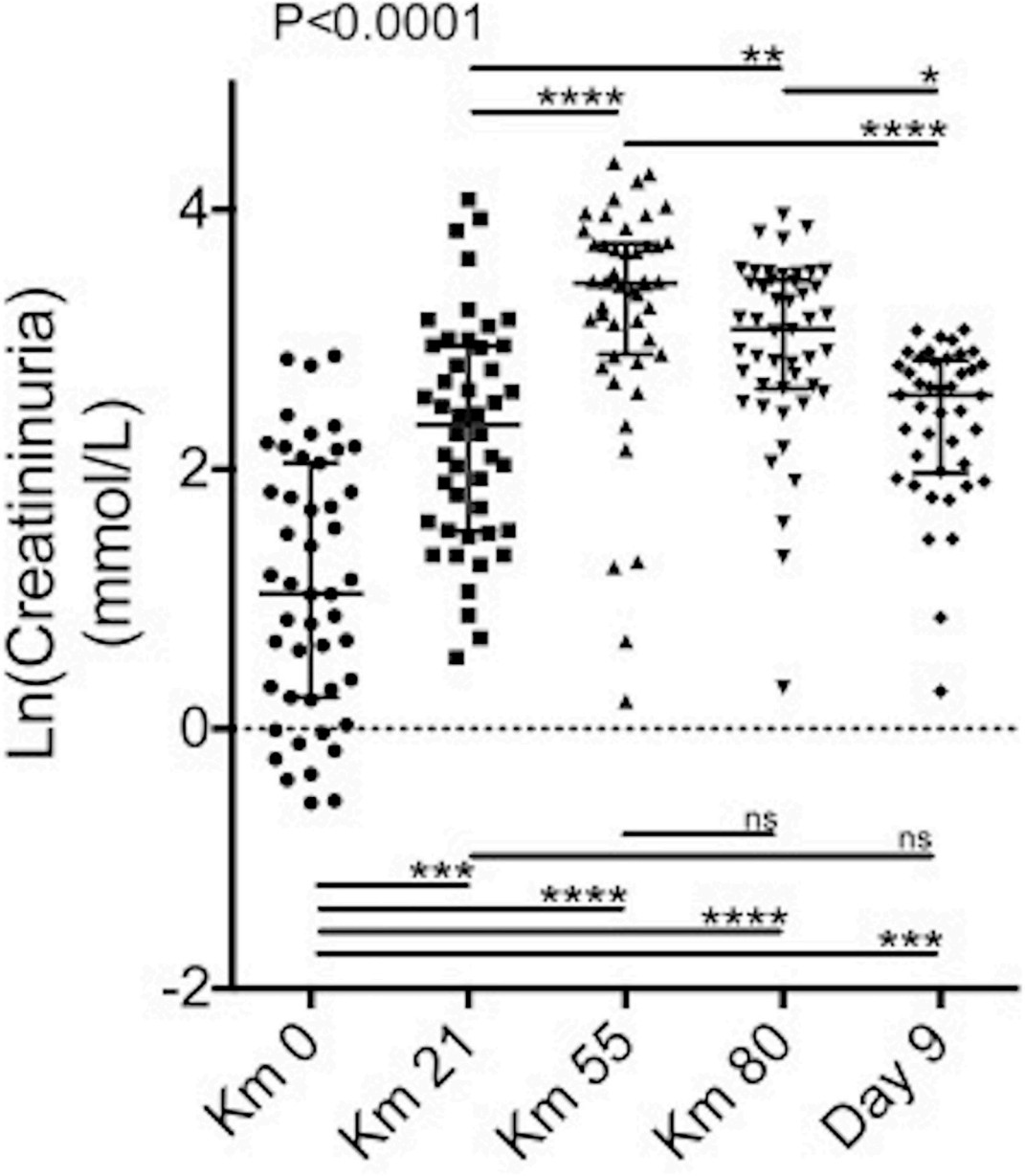

**Fig 1. Longitudinal evolution of urine creatinine.** Scatter plots of the log (natural)-transformed urine creatinine levels of 47 patients at km 0, km 21, km 55, km 80 and at day 9 after the race. p values are based on the Kruskal-Wallis test. Stars depict pairwise group comparisons by means of Dunn's post-test (*p<0.05; **p<0.01; ***p<0.001; ****p<0.0001).

death, rhabdomyolysis, electrolyte disorders and negative impact on kidney function were reported [16–20]. Nevertheless, the temporal variations of kidney injury biomarkers during the race was not previously studied. In this study, results from a few participants evidenced an AKI according to KDIGO AKI score [21, 22]. Nevertheless, after adjustment on urine dilution, no significant dramatic changes remained. Furthermore, all AKI biomarker variations fully disappeared on day 9.

Acute dehydration is a well-known risk factor for prerenal azotemia that can lead to true organ damage by acute tubular necrosis if profound and sustained. The lack of biological

**Table 3. Urinary biomarker levels in 47 runners.** Data are expressed as mean ± standard deviation (SD) and minimal–maximal values (Min–Max).

| Parameter | km 0 (baseline) | km 21 | km 53 | km 80 (arrival) | Day 9 |
|---|---|---|---|---|---|
| **NGAL, pg/mL** | | | | | |
| Mean ± SD | 15164 ± 19919 | 39330 ± 34848 | 75684 ± 65368 | 38447 ± 41023 | 24940 ± 28164 |
| Median | 7008 (95878) | 30090 (142006) | 57859 (313305) | 24702 (188124) | 15547 (132052) |
| Min–Max | 2235–98113 | 2500–144506 | 2500–315805 | 2500–190624 | 2045–134097 |
| **NGAL:Cr, pg/mmol** | | | | | |
| Mean ± SD | 4433 ± 5287 | 3735 ± 3117 | 3776 ± 7413 | 1713 ± 1968 | 2349 ± 2372 |
| Median (IQR) | 3302 (3471) | 2771 (3004) | 2196 (2345) | 1208 (1240.7) | 1643 (2178.8) |
| Min–Max | 257.7–31077 | 130.6–13686 | 79.47–49999 | 149.6–12195 | 143.4–8864 |
| **KIM1, pg/mL** | | | | | |
| Mean ± SD | 171.6 ± 177.5 | 778.2 ± 1156 | 2204 ± 2222 | 1875 ± 2212 | 672.7 ± 551.7 |
| Median (IQR) | 80.00 (161) | 299.5 (822) | 1433 (1927.9) | 1261 (2131.4) | 475.0 (747.2) |
| Min–Max | 80.00–788.2 | 55.95–5776 | 80.00–13947 | 80.00–13688 | 80.00–2232 |
| **KIM1:Cr, pg/mmol** | | | | | |
| Mean ± SD | 61.07 ± 74.92 | 47.10 ± 35.66 | 107.1 ± 176.6 | 74.07 ± 67.45 | 51.03 ± 29.27 |
| Median (IQR) | 40.56 (51.37) | 41.73 (47.87) | 53.79 (49.35) | 59.76 (55.91) | 43.81 (31.84) |
| Min—Max | 8.248–516.1 | 5.027–184.1 | 9.660–962.8 | 2.668–461.0 | 15.98–147.8 |

NGAL = neutrophil gelatinase associated lipocalin, NGAL:Cr = NGAL:creatinine ratio (NGAL:Cr, expressed in pg protein/mmoL urine creatinine), KIM1 = Kidney Injury Molecule-1, KIM1:Cr = KIM1:screatinine ratio (KIM1:Cr, expressed in pg protein/mmoL urine creatinine).

parameters of dehydration observed in this study supports the conclusion that an ultra-distance race does not induce a short-term major kidney injury.

Additional factors related to ultra-distance exercise-practice may also trigger AKI, including hypoxemia, production of reactive oxygen species [22], and pro-inflammatory cytokine production [23]. For instance, ultra-distance races have been shown to increase the accumulation of nitrotirosine and protein carbamyl in serum and urine, thereby suggesting the induction of oxidative stress induced by extreme physical exercise [22].

Questioning the temporal variations of AKI occurrence and resolution after an ultra-distance race is not a futile question. For decades, AKI has been considered as a benign pathological condition that usually recovers "ad integrum" with no definitive kidney damage (9). However, epidemiological and experimental studies recently demonstrated a true interconnection of AKI and chronic kidney disease (CKD), and it is now considered that repeated episodes of AKI, including mild cases, may induce CKD over the long term [9, 24]. Observational studies have even demonstrated that AKI leads to new CKD, progression of existing CKD, increasing long-term risk of end-stage renal disease and increased mortality [9, 11, 14, 25]. The question as to whether ultra-distance races might trigger mild and repeated AKI episodes has therefore important implications for CKD prevention.

The high level of urine creatinine concentration, observed during the race, doesn't necessarily reflect the high level of urine concentration but may simply be reflective of the higher blood creatinine level due to muscle damage [26]. Specifically, in the context of an ultra-distance race, increased serum creatinine may reflect pure prerenal azotemia with no immediate consequence on renal tissue integrity. We cannot know how much of the rise in serum creatinine observed during the race is due to a true creatinine clearance decrease or to an increase in muscle release of creatinine. In this study, we used KIM1 and NGAL urinary levels, two well-known biomarkers of AKI, to assess renal function, because traditional blood (creatinine and blood urea nitrogen concentrations) and urine markers of kidney injury (casts, fractional

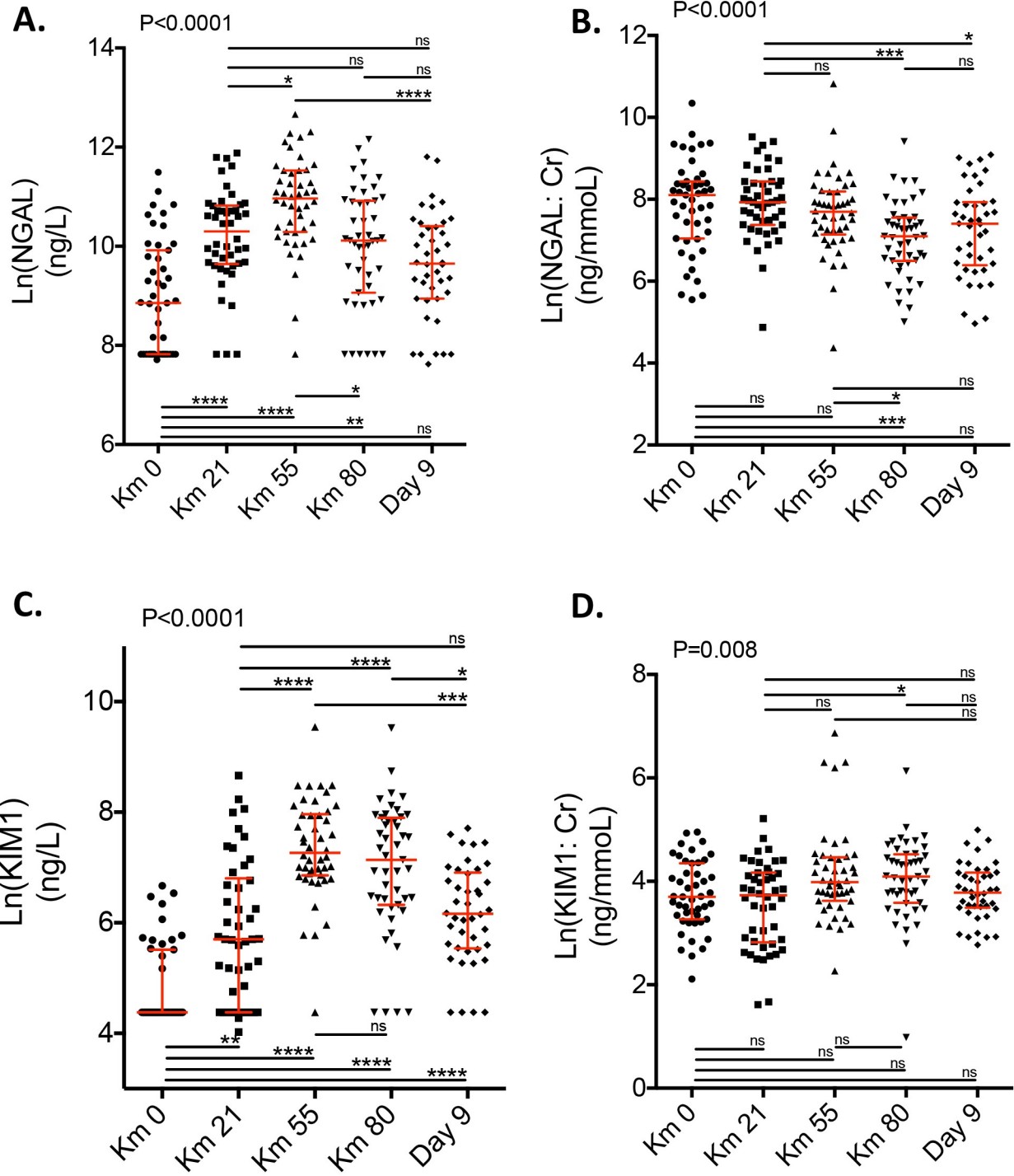

**Fig 2. Longitudinal evolution of NGAL and KIM1.** Scatter plots of the log (natural)-transformed values of NGAL (A. and B.) and KIM1 (C. and D.) levels of 47 patients at km 0, km 21, km 55, km 80 and at day 9 after the race. Results are normalized (B. and D.) or not (A. and C.) by urine creatinine levels. P values are based on the Kruskal-Wallis test. Stars depict pairwise group comparisons by means of Dunn's post-test (*p<0.05; **p<0.01; ***p<0.001; ****p<0.0001). NGAL = neutrophil gelatinase associated lipocalin, KIM1 = Kidney Injury Molecule-1.

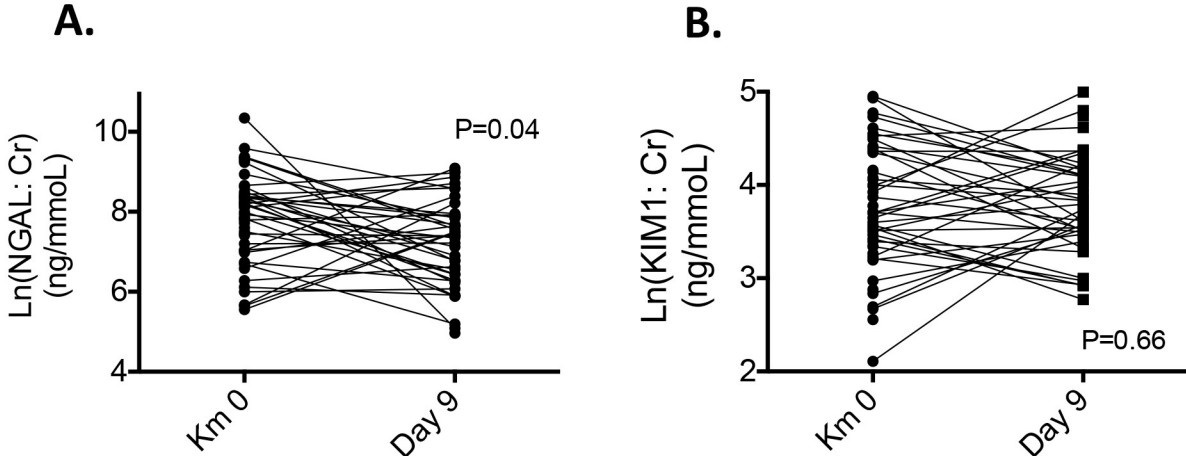

**Fig 3. Paired analysis of NGAL:Cr and KIM1:Cr ratios between Km 0 and Day 9 after the race.** P values are based on the Wilcoxon matched-pairs signed rank test. NGAL (pg/mL) and NGAL:creatinine ratio (NGAL:Cr, expressed in pg protein/mmoL urine creatinine), KIM1:creatinine ratio (KIM1:Cr, expressed in pg protein/mmoL urine creatinine).

excretion of sodium, urinary concentration ability) are insensitive and nonspecific [27] contrary to muddy brown urinary casts indicative of tubular injury.

Mc Cullough et al. were the first to study urinary KIM1 and NGAL levels after marathon reporting a transiently increased level of these biomarkers during the race [20]. Lippi et al. [28] confirmed the acute variations of serum creatinine and urinary NGAL occurring after an ultra-distance race. In our study, we didn't retrieve significant urinary NGAL variations, because we assessed the urinary NGAL value normalized to the urinary creatinine level to take into account the potential confounding effect on acute renal assessment. Due to the highly variable dilution status of the urine samples related to the exercise, urinary AKI biomarker levels need to be cautiously interpreted [29]. Indeed, our findings that urinary creatinine, used as a marker of urine dilution, is highly variable during and after the race should be considered for future urine biomarker interpretation.

Conversely, urine creatinine dramatically increased during the race suggesting extracellular dehydration and antidiuresis. While changes during and after the race of NGAL and KIM1 were quite impressive in our study, the variability over time was profoundly reduced after normalization by urine creatinine. After correction for urine dilution status on urine creatinine level, NGAL and KIM1 urine levels were not significantly different between baseline (km 0) and km 80, suggesting the limited impact of an ultra-distance race on renal tubular cell integrity.

NGAL, a proinflammatory mediator protein of the innate immune system, is a helpful tool to differentiate prerenal and intrinsic AKI [30–34]. Since, in our study, we observed no difference between baseline and other points (during and on day 9 after the running), we can hypothesize that AKI related to exercise practice is prerenal. Thus, the limited impact on kidney function observed herein may partly be explained by the free access to food and water in the study design. Nevertheless, beside extracellular volume depletion, we cannot exclude the contribution from other mechanisms to explain prerenal AKI. Among these, we can cite such the effects of the sympathetic nervous system on renal blood flow, and/or antidiuretic effect of vasopressin released in response to stress and/or pain, and/or cardiac dysfunction.

Beyond the differential diagnosis of AKI, urinary NGAL has a prognostic value to predict adverse outcome in both adult and children [6–11]. Urinary NGAL predicts the severity of

AKI after cardiac surgery and long-term renal outcome after ICU stay [35, 36]. Urinary NGAL and urinary KIM-1 have good prognostic value for AKI [35–40]. Conversely, for CKD, they don't provide robust prognostic information on renal function loss or because both biomarkers are sensitive to detect acute tubular injury but not tubular atrophy [41].

In this study, some limitations deserve consideration. First, this is a single centre study with a small sample size. Second, we did not obtain blood sample for all participants. Third, food and water intakes were not restricted and controlled during the race. Fourth, physical activity and diet in the short term before and after the race were not controlled; thus, we cannot rule out their impact on serum and urinary biomarkers levels. Fifth, we made the assumption, that the runners were healthy, but no comparison with a healthy control population was possible and performed to ascertain it.

Despite of these limitations, this first study with a longitudinal follow-up of blood and urine parameters suggests the restricted impact on short term of an ultra-distance race on the kidney function by combining blood and urinary parameters.

In conclusion, our study is the first to assess the temporal variations of the renal function during an ultra-distance race based on serially collected blood and urine samples with a follow-up 9 days after the end of the race. Half of the subjects experienced acute and transient increase in the serum creatinine levels, but urinary biomarkers of AKI displayed only limited changes over time, thereby suggesting that ultra-trail has no impact on acute kidney function.

## Acknowledgments

We thank the Ecotrail de Paris Ile de France© race organizers and all runners involved in the study.

## Author Contributions

**Conceptualization:** Romain Jouffroy, Nicolas Mansencal.

**Data curation:** Romain Jouffroy, Nicolas Mansencal.

**Formal analysis:** Xavier Lebreton, Dany Anglicheau.

**Investigation:** Romain Jouffroy.

**Supervision:** Romain Jouffroy.

**Validation:** Romain Jouffroy, Dany Anglicheau.

**Visualization:** Romain Jouffroy, Dany Anglicheau.

**Writing – original draft:** Romain Jouffroy, Nicolas Mansencal, Dany Anglicheau.

**Writing – review & editing:** Romain Jouffroy, Nicolas Mansencal, Dany Anglicheau.

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
