## [Decision Letter · Decision Letter 0]

23 Jul 2019

PONE-D-19-17069

Acute kidney injury during an ultra-distance race

PLOS ONE

Dear Dr Jouffroy,

Thank you for submitting your manuscript to PLOS ONE. After careful consideration, we feel that it has merit but does not fully meet PLOS ONE’s publication criteria as it currently stands. Therefore, we invite you to submit a revised version of the manuscript that addresses the points raised during the review process.

ACADEMIC EDITOR: 

Both reviewers raised important issues that should be carefully addressed to improve the quality of the manuscript.

We would appreciate receiving your revised manuscript by Sep 06 2019 11:59PM. To enhance the reproducibility of your results, we recommend that if applicable you deposit your laboratory protocols in protocols.io, where a protocol can be assigned its own identifier (DOI) such that it can be cited independently in the future. For instructions see: http://journals.plos.org/plosone/s/submission-guidelines#loc-laboratory-protocols

We look forward to receiving your revised manuscript.

Kind regards,

Paolo Cravedi

Academic Editor

PLOS ONE

Journal Requirements:

1. Please ensure that all statements in the introduction and discussion sections are accurate and adequately supported: e.g. "Physical activity also prevents lifestyle-related metabolic diseases (3) and results in few cancers occurrence (4).

2. Thank you for including your funding statement; "not applicable"

Please provide an amended Funding Statement that declares *all* the funding or sources of support received during this specific study (whether external or internal to your organization) as detailed online in our guide for authors at http://journals.plos.org/plosone/s/submit-now.  

Please state what role the funders took in the study.  If any authors received a salary from any of your funders, please state which authors and which funder. If the funders had no role, please state: "The funders had no role in study design, data collection and analysis, decision to publish, or preparation of the manuscript."

Reviewers' comments:

Reviewer's Responses to Questions

**Comments to the Author**

1. Is the manuscript technically sound, and do the data support the conclusions?

Reviewer #1: Partly

Reviewer #2: Yes

2. Has the statistical analysis been performed appropriately and rigorously? 

Reviewer #1: I Don't Know

Reviewer #2: Yes

3. Have the authors made all data underlying the findings in their manuscript fully available?

Reviewer #1: Yes

Reviewer #2: Yes

4. Is the manuscript presented in an intelligible fashion and written in standard English?

Reviewer #1: Yes

Reviewer #2: Yes

5. Review Comments to the Author

Reviewer #1: The authors report on the temporal changes in kidney injury biomarkers before, during, and after an 80km ultra-distance race. The study was designed and conducted appropriately. The manuscript has potential and can benefit from some improved clarity with the results and associated conclusions. Additionally, the focus of the introduction and discussion can be expanded. A large focus of the introduction and discussion involves the AKI to CKD transition. Although this is a good point to make, the study had short term follow-up. The authors should expand these sections to include other exercise related AKIs, such as from “regular” marathons and spin classes. Several studies and numerous case reports have been published within the last few years, notably, one published in PLoS ONE. Additionally, Mansour et al. AJKD 2017, is an impactful study and should be discussed. The authors should add a discussion on the strengths and limitations of KIM-1 and NGAL, including the differences between blood and urine measurement.

Also, the novelty the authors make regarding their study of an ultra-distance race rather than a “regular” marathon is probably not as clinically significant as the authors make it out to be. This should be adjusted throughout the manuscript including the last two paragraphs of the discussion.

Minor comments:

The in-text tables and figure legends (pages 8-10) are confusing to follow and should be moved to the proper section according to the journal’s author instructions. Also, the landscape formatting on the later pages needs to be fixed.

Introduction

Page 4, line 3 – clarify what the 480,000 runners refers to specifically. Did 480,000 runners complete a marathon in that one year or in 2001, have 480,000 runners completed a marathon at some point in their life?

Methods

Page 5, line 8 – clarify what medical disease refers to. Is this specifically referring to chronic diseases? What about occasional headache or infections? Only systemic illness?

Page 5, lines 9-10 – The sentence needs to be rewritten. I believe the authors intent to say that the participants had free access to food and water during and after the race. “ad libitum”

The study protocol was approved in 2014, was this the same year the race occurred? The authors should mention the year of the race. At least they mentioned the weather that day.

Results

Page 8 line 12 – Clarify the sentence to specify mean time of the race is referring to “mean race completion time”

Table 1 – what is “Trial experience”?

Page 10, line 18 – elevated urine creatinine does not necessarily reflect a “high level of urine concentration”, which is determined from the measurement of specific gravity or osmolarity. The high urine creatinine may simply be reflective of the higher blood creatinine level. Measurement of urinary osmolarity and electrolytes in stored samples would provide additional insight into kidney function during and after the race.

How much of the rise in serum creatinine during the race is really a true reduction in CrCl vs. and increase in muscle release of creatinine? The authors should comment on this.

It would be interesting to see the breakdown of the kidney injury biomarkers by KDIGO AKI stage. Were the biomarkers more elevated in these participants?

Discussion

Page 14, line 4 – Although the authors comment on AKI to CKD transition in other sections, they should specify here safe for “short term” kidney function as their study followed patients up to 9 days after the race.

Page 14, lines 6-7 – This sentence is not accurate, The impact of long-distance running on kidney function has been studied. As mentioned earlier, the Mansour AJKD paper, and others.

Since the authors talk about acute dehydration leading to ATN in the discussion, they should mention that the clinical parameters in table 2 do reflect any clinical evidence of dehydration and supports their finding of no major kidney injury.

Page 15, line 7 – remove the word “the”

Page 15, line 8 – maybe hyaline and waxy casts are nonspecific but muddy brown urinary casts are not nonspecific and are indicative of tubular injury.

Page 15, lines 14-15 – the assumption that urine was highly diluted before the race is incorrect. There is no evidence to support this claim. Additionally, what does “huge fluid consumption” actually mean?

Reviewer #2: The Authors in their interesting work show that transient increase in creatinine is not accompanied to parallel increase of two urinary biomarkers of acute kidney injury (uNGAL/uCr and uKIM 1/uCr) (ultra-distance race) in trained athletes during intense physical effort.

Major criticisms

-The Authors show that uNGAL/uCr significantly decreases from baseline to day 9 after the physical effort, however in the discussion they state that “…all biomarker variation fully disappeared on day 9...” and “…urine levels of NGAL..were similar at 80 km and at baseline after correction for urine dilution status”. I believe that the decrease of uNGAL/uCr during the physical effort and at day 9 after the race deserves an appropriate discussion. Do the Authors have any data regarding the physical activity/diet in the short term before (ideally 7-10 days) and after the race ? Furthermore it would be interesting to compare baseline athletes’ uNGAL/uCr with healthy control population . Finally it has been previously reported that uNGAL/uCr increases in athletes after 60 kg ultramarathon ( Clin Chem Lab Med. 2012 Feb 14;50(9):1585-9. doi: 10.1515/cclm-2011-0954), please discuss (different population/analysis method?).

Minor criticism

- eGFR should be reported in Table 2

- I would suggest to report the number of subjects with blood test in Table 2 legend (as done by the Authors in Table 3 legend)

6. PLOS authors have the option to publish the peer review history of their article (what does this mean?). If published, this will include your full peer review and any attached files.

Reviewer #1: Yes: Joshua Rein

Reviewer #2: No

---

## [Author Response · Author response to Decision Letter 0]

7 Aug 2019

Dear Editor,

We thank the reviewers for their work and their interesting remarks.

Please consider the revised version of our manuscript.

Please find below the answers to the reviewers’ comments.

All substantial modifications appear in red and bold in the revised manuscript.

Sincerely yours.

Romain Jouffroy

Reviewer's Responses to Questions

Comments to the Author

Reviewer #1: The authors report on the temporal changes in kidney injury biomarkers before, during, and after an 80km ultra-distance race. The study was designed and conducted appropriately. The manuscript has potential and can benefit from some improved clarity with the results and associated conclusions. Additionally, the focus of the introduction and discussion can be expanded. A large focus of the introduction and discussion involves the AKI to CKD transition. Although this is a good point to make, the study had short term follow-up. The authors should expand these sections to include other exercise related AKIs, such as from “regular” marathons and spin classes. Several studies and numerous case reports have been published within the last few years, notably, one published in PLoS ONE. Additionally, Mansour et al. AJKD 2017, is an impactful study and should be discussed. The authors should add a discussion on the strengths and limitations of KIM-1 and NGAL, including the differences between blood and urine measurement.

Answer: We thank the reviewer for the remark. In the revised version of the manuscript, the introduction and discussion sections were expended. Nevertheless, we believe that long-distance running is different from regular marathon running because the speed is quite lower, and most of races take place on trails with denivelation contrary to regular marathon on road. Another originality of our study is the kinetic of AKI biomarkers contrary to previous studies with different methodology. In previous studies, the impact of running is evaluated comparing before and after running biomarkers values. 

Also, the novelty the authors make regarding their study of an ultra-distance race rather than a “regular” marathon is probably not as clinically significant as the authors make it out to be. This should be adjusted throughout the manuscript including the last two paragraphs of the discussion.

Answer: We thank the reviewer for this interesting suggestion. In the revised version of the manuscript, manuscript and the last two paragraphs of the discussion were adjusted in order to take into account the suggestion.

Minor comments:

The in-text tables and figure legends (pages 8-10) are confusing to follow and should be moved to the proper section according to the journal’s author instructions. Also, the landscape formatting on the later pages needs to be fixed.

Answer: We agree with this remark. In the revised version of the text, the landscape formatting was modified in order to accord with journal’s author instructions.

Introduction

Page 4, line 3 – clarify what the 480,000 runners refers to specifically. Did 480,000 runners complete a marathon in that one year or in 2001, have 480,000 runners completed a marathon at some point in their life?

Answer: We agree with this remark. In the revised version, the sentence was modified in order to avoid confusion.

Methods

Page 5, line 8 – clarify what medical disease refers to. Is this specifically referring to chronic diseases? What about occasional headache or infections? Only systemic illness?

Answer: We fully agree with this comment. The medical disease refers to chronic diseases. 

A sentence has been added in the new version of the manuscript to clarify this point.

Page 5, lines 9-10 – The sentence needs to be rewritten. I believe the authors intent to say that the participants had free access to food and water during and after the race. “ad libitum”.

Answer: We thank the reviewer for this remark. In the revised version of the manuscript, the sentence was rewritten. 

The study protocol was approved in 2014, was this the same year the race occurred? 

The authors should mention the year of the race. At least they mentioned the weather that day.

Answer: We thank the reviewer for the remark and the suggestion. In the revised version of the manuscript, the year of the race and the day race weather were added. 

Results

Page 8 line 12 – Clarify the sentence to specify mean time of the race is referring to “mean race completion time”

Answer: We thank the reviewer for this interesting suggestion. In the revised version of the manuscript, the sentence was rewritten.

Table 1 – what is “Trial experience”?

Answer: We thank the reviewer for this remark. In the revised version of the text, the term was changed for “seniority in running practice”. 

Page 10, line 18 – elevated urine creatinine does not necessarily reflect a “high level of urine concentration”, which is determined from the measurement of specific gravity or osmolarity. The high urine creatinine may simply be reflective of the higher blood creatinine level. Measurement of urinary osmolarity and electrolytes in stored samples would provide additional insight into kidney function during and after the race.

Answer: We fully agree with this remark and thank the reviewer. In the revised version of the manuscript, the sentence was deleted in the results section and a sentence added in the discussion.

How much of the rise in serum creatinine during the race is really a true reduction in CrCl vs. and increase in muscle release of creatinine? The authors should comment on this.

Answer: We thank the reviewer for this very interesting remark. A sentence was inserted in the discussion to emphasize this very important point.

It would be interesting to see the breakdown of the kidney injury biomarkers by KDIGO AKI stage. Were the biomarkers more elevated in these participants?

Answer: We thank the reviewer for this interesting suggestion. A sentence was added in the results section to describe the kinetics of kidney injury biomarkers by KDIGO AKI stage.

Discussion

Page 14, line 4 – Although the authors comment on AKI to CKD transition in other sections, they should specify here safe for “short term” kidney function as their study followed patients up to 9 days after the race.

Answer: We fully agree with this suggestion and thank the reviewer. In the revised version of the manuscript, the sentence was modified in the abstract and manuscript conclusions. 

Page 14, lines 6-7 – This sentence is not accurate, the impact of long-distance running on kidney function has been studied. As mentioned earlier, the Mansour AJKD paper, and others.

Answer: We thank the reviewer for this remark. The sentence was rewritten in order to underline the originality of our study showing the kinetic of kidney injury biomarkers. 

Since the authors talk about acute dehydration leading to ATN in the discussion, they should mention that the clinical parameters in table 2 do reflect any clinical evidence of dehydration and supports their finding of no major kidney injury.

Answer: We thank the reviewer for this suggestion. A sentence was added in the results section to describe the kinetics of kidney injury biomarkers by KDIGO AKI stage.

Page 15, line 7 – remove the word “the”

Answer: We thank the reviewer for this remark. In the revised version of the manuscript, the term “the” was deleted.

Page 15, line 8 – maybe hyaline and waxy casts are nonspecific but muddy brown urinary casts are not nonspecific and are indicative of tubular injury.

Answer: We agree and thank the reviewer for this remark. The sentence was rewritten in order to be more accurate. 

Page 15, lines 14-15 – the assumption that urine was highly diluted before the race is incorrect. There is no evidence to support this claim. Additionally, what does “huge fluid consumption” actually mean?

Answer: We agree with this remark and thank the reviewer. In the revised version of the manuscript, the sentence was deleted. By “huge fluid consumption”, we wanted to say that a possible explanation was the increase fluid intake before the race by most of runners.

Reviewer #2: The Authors in their interesting work show that transient increase in creatinine is not accompanied to parallel increase of two urinary biomarkers of acute kidney injury (uNGAL/uCr and uKIM 1/uCr) (ultra-distance race) in trained athletes during intense physical effort.

We thank the reviewer for his comment.

Major criticisms

-The Authors show that uNGAL/uCr significantly decreases from baseline to day 9 after the physical effort, however in the discussion they state that “…all biomarker variation fully disappeared on day 9...” and “…urine levels of NGAL were similar at 80 km and at baseline after correction for urine dilution status”. 

Answer: We thank the reviewer for the remark. Figure 2B shows no difference between uNGAL/uCr between baseline (km 0) and day 9 whereas a significant difference was observed between baseline and the end of the race (km 80). The sentence, in the discussion, was rewritten in order to avoid confusion; we fully agree that “similar” and “not different” are not, statistically, synonymous.

I believe that the decrease of uNGAL/uCr during the physical effort and at day 9 after the race deserves an appropriate discussion. Do the Authors have any data regarding the physical activity/diet in the short term before (ideally 7-10 days) and after the race? 

Answer: We thank the reviewer and fully agree with this comment. 

We don’t have any data regarding the physical activity/diet in the short term before and after the race. An add was inserted in the revised version of the manuscript to emphasize the important impact of this limitation. 

Furthermore, it would be interesting to compare baseline athletes’ uNGAL/uCr with healthy control population. 

Answer: We thank the reviewer for this very interesting suggestion. Unfortunately, we don’t have data to compare baseline athletes’ uNGAL/uCr and healthy control population. A limitation was inserted in the revised version of the manuscript to underline this point. 

Finally it has been previously reported that uNGAL/uCr increases in athletes after 60 kg ultramarathon (Clin Chem Lab Med. 2012 Feb 14;50(9):1585-9. doi: 10.1515/cclm-2011-0954), please discuss (different population/analysis method?).

Answer: We thank the reviewer for this reference. The reference has been added and potential explanations for the differences in studies results inserted in the discussion. 

Minor criticism

- eGFR should be reported in Table 2

Answer: We thank the reviewer for this suggestion. In the revised version of the manuscript, eGFR (mean +/- SD) was added in Table 2.

- I would suggest to report the number of subjects with blood test in Table 2 legend (as done by the Authors in Table 3 legend)

Answer: We thank the reviewer for this suggestion. In the revised version of the manuscript, the number of subjects with blood test was added in Table 2 legend.

---

## [Decision Letter · Decision Letter 1]

20 Aug 2019

PONE-D-19-17069R1

Acute kidney injury during an ultra-distance race

PLOS ONE

Dear Dr Jouffroy,

Thank you for submitting your manuscript to PLOS ONE. After careful consideration, we feel that it has merit but does not fully meet PLOS ONE’s publication criteria as it currently stands. Therefore, we invite you to submit a revised version of the manuscript that addresses the final minor points raised during by Reviewer 1.

We would appreciate receiving your revised manuscript by Oct 04 2019 11:59PM. To enhance the reproducibility of your results, we recommend that if applicable you deposit your laboratory protocols in protocols.io, where a protocol can be assigned its own identifier (DOI) such that it can be cited independently in the future. For instructions see: http://journals.plos.org/plosone/s/submission-guidelines#loc-laboratory-protocols

We look forward to receiving your revised manuscript.

Kind regards,

Paolo Cravedi

Academic Editor

PLOS ONE

Reviewers' comments:

Reviewer's Responses to Questions

**Comments to the Author**

1. If the authors have adequately addressed your comments raised in a previous round of review and you feel that this manuscript is now acceptable for publication, you may indicate that here to bypass the “Comments to the Author” section, enter your conflict of interest statement in the “Confidential to Editor” section, and submit your "Accept" recommendation.

Reviewer #1: (No Response)

Reviewer #2: All comments have been addressed

2. Is the manuscript technically sound, and do the data support the conclusions?

Reviewer #1: Yes

Reviewer #2: Yes

3. Has the statistical analysis been performed appropriately and rigorously? 

Reviewer #1: I Don't Know

Reviewer #2: Yes

4. Have the authors made all data underlying the findings in their manuscript fully available?

Reviewer #1: Yes

Reviewer #2: Yes

5. Is the manuscript presented in an intelligible fashion and written in standard English?

Reviewer #1: No

Reviewer #2: Yes

6. Review Comments to the Author

Reviewer #1: I thank the authors for their revisions that adequately addressed my major concerns. I have several additional comments/suggestions to further improve the manuscript for publication.

I appreciate the authors’ clarification regarding the novelty of their study in comparison to prior studies. The authors should add a sentence or two introducing what trail running is as explained in their response to my first comment. They can include aspects such as terrain, running surface, flat vs. hills, carry extra weight, and make a comparison with marathons to help clarify and explain differences in the races.

Throughout the manuscript, the authors should emphasize “ultra-distance” when referring to the race rather than “long-distance” to distinguish from marathons.

Authors use the words running where in some sentences, race would be the more appropriate word. The sentences can be reworded so that running also fits if the authors prefer to use that word. Page 3 lines 4 and 5 – “biomarker injury while running” or “biomarker injury during the race”

Temporal seems to be a more appropriate word than kinetic when describing changes in biomarkers over time. Kinetic refers more to the changes in the rate of something. The authors can consider calculating rates of change at each time point but I don’t think that would be of any clinical significance.

As some of the authors have previously published on changes in cardiac function during ultradistance racing, do the authors think the reduction in eGFR can be partially explained by cardiac strain? Also, the authors state in the discussion on page 18 line 22 that AKI may be prerenal. Besides extracellular volume depletion, it may be worth mentioning other potential mechanisms such effects on renal blood flow by the sympathetic nervous system, and antidiuresis from vasopressin release in response to stress/pain. May be worth commenting on to build the discussion but not critical.

Page 5 line 3 - As the authors discuss recent trends in increasing race distance in several parts of the manuscript, they should consider using a more recent reference than the one used from 2001.

Page 5 line 2 and line 10 – fix the use of two vs. 2

Page 5 line 24 – clarify that “before and after” is referring to the race

Page 6 line 3 – blood and urine samples were also collected before the race.

Did participants exclusively drink water or could some have consumed sports drinks? Might be worth discussing the association with kidney injury and rehydration with sugar sweetened beverages if participants did in fact have access to it.

It is more appropriate to use the word “chloride” rather than “chlorine” in the text and tables.

Page 12 lines 9-10 – Myoglobin increased from 34 to 2761, if not statistically significant, this increase may possibly be of some clinical significance. Although not massively elevated, do the authors believe this elevation could have contributed to any reduction in eGFR?

Page 14 lines 12-16 – Awkward sentence structure. These sentences would benefit from English editing.

Page 18 line 23 - If the hypothesis is that free access to food and water limited the kidney injury, it is safe to say on page 18 lines 12-13 that suggesting dehydration and antidiuresis?

Page 19 line 21 – ultraendurance races do in fact have considerable effects to kidney function such as, as the authors state, antidiuresis. However, it would be more accurate to state that ultra-endurance races specifically have no impact on acute kidney injury.

Reviewer #2: (No Response)

7. PLOS authors have the option to publish the peer review history of their article (what does this mean?). If published, this will include your full peer review and any attached files.

Reviewer #1: No

Reviewer #2: No

---

## [Author Response · Author response to Decision Letter 1]

29 Aug 2019

Dear Editor,

We thank the reviewers for their work and their interesting remarks.

Please consider the revised version of our manuscript.

Please find below the answers to the reviewers’ comments.

In the revised manuscript, all substantial modifications appear in red and bold whereas previous modifications remain in black and bold.

Sincerely yours.

Romain Jouffroy

Responses to Reviewers' comments:

Comments to the Author

1. If the authors have adequately addressed your comments raised in a previous round of review and you feel that this manuscript is now acceptable for publication, you may indicate that here to bypass the “Comments to the Author” section, enter your conflict of interest statement in the “Confidential to Editor” section, and submit your "Accept" recommendation.

Reviewer #1: (No Response)

Reviewer #2: All comments have been addressed

2. Is the manuscript technically sound, and do the data support the conclusions?

Reviewer #1: Yes

Reviewer #2: Yes

3. Has the statistical analysis been performed appropriately and rigorously? 

Reviewer #1: I Don't Know

Reviewer #2: Yes

4. Have the authors made all data underlying the findings in their manuscript fully available?

Reviewer #1: Yes

Reviewer #2: Yes

5. Is the manuscript presented in an intelligible fashion and written in standard English?

Reviewer #1: No

Reviewer #2: Yes 

6. Review Comments to the Author

Reviewer #1: I thank the authors for their revisions that adequately addressed my major concerns. I have several additional comments/suggestions to further improve the manuscript for publication.

Answer: We thank the reviewer for his work and new comments/suggestions to further improve our manuscript. 

I appreciate the authors’ clarification regarding the novelty of their study in comparison to prior studies. The authors should add a sentence or two introducing what trail running is as explained in their response to my first comment. They can include aspects such as terrain, running surface, flat vs. hills, carry extra weight, and make a comparison with marathons to help clarify and explain differences in the races.

Answer: We thank you for this suggestion. In the revised manuscript, three sentences have been added to better explain to the readers what trail running is in comparison with marathons.

Throughout the manuscript, the authors should emphasize “ultra-distance” when referring to the race rather than “long-distance” to distinguish from marathons.

Answer: We thank the reviewer for this suggestion. In the revised manuscript, the term “long-distance” was replaced by “ultra-distance”. 

Authors use the words running where in some sentences, race would be the more appropriate word. The sentences can be reworded so that running also fits if the authors prefer to use that word. Page 3 lines 4 and 5 – “biomarker injury while running” or “biomarker injury during the race”.

Answer: We thank the reviewer for this suggestion. In the revised manuscript, the term running was replaced by race.

Temporal seems to be a more appropriate word than kinetic when describing changes in biomarkers over time. Kinetic refers more to the changes in the rate of something. 

Answer: We thank the reviewer for this interesting remark. In the revised manuscript, the word “kinetic” running was replaced by “temporal”.

The authors can consider calculating rates of change at each time point but I don’t think that would be of any clinical significance.

Answer: We fully agree with the reviewer. As underlined, the rate of change does not provide more clinical information in this study. 

As some of the authors have previously published on changes in cardiac function during ultradistance racing, do the authors think the reduction in eGFR can be partially explained by cardiac strain? 

Also, the authors state in the discussion on page 18 line 22 that AKI may be prerenal. Besides extracellular volume depletion, it may be worth mentioning other potential mechanisms such effects on renal blood flow by the sympathetic nervous system, and antidiuresis from vasopressin release in response to stress/pain. May be worth commenting on to build the discussion but not critical.

Answer: We fully agree with this very interesting remark. A sentence has been inserted in the discussion in order to take into account the remark. 

Page 5 line 3 - As the authors discuss recent trends in increasing race distance in several parts of the manuscript, they should consider using a more recent reference than the one used from 2001.

Answer: We thank the reviewer for this suggestion. In the revised manuscript, a more recent reference replaced the one of 2001. Consequently, the sentence was modified. 

Page 5 line 2 and line 10 – fix the use of two vs. 2

Answer: We thank the reviewer. The modification appears in the revised version of the manuscript. 

Page 5 line 24 – clarify that “before and after” is referring to the race

Answer: We thank the reviewer. The modification appears in the revised version of the manuscript.

Page 6 line 3 – blood and urine samples were also collected before the race.

Answer: We fully agree with the reviewer. The term “before” was added in the sentence.

Did participants exclusively drink water or could some have consumed sports drinks? Might be worth discussing the association with kidney injury and rehydration with sugar sweetened beverages if participants did in fact have access to it.

Answer: We thank the reviewer for this suggestion. The participants drink only water during the race. A sentence was inserted in the revised version of the manuscript to clarify this point.

It is more appropriate to use the word “chloride” rather than “chlorine” in the text and tables.

Answer: We thank the reviewer. The modification appears in the revised version of the manuscript.

Page 12 lines 9-10 – Myoglobin increased from 34 to 2761, if not statistically significant, this increase may possibly be of some clinical significance. Although not massively elevated, do the authors believe this elevation could have contributed to any reduction in eGFR?

Answer: We fully agree with the author that the myoglobin elevation at the end of the race vs start could partly explain the reduction in eGFR. 

Page 14 lines 12-16 – Awkward sentence structure. These sentences would benefit from English editing.

Answer: We thank the reviewer. The sentences were edited in the new version of the manuscript. 

Page 18 line 23 - If the hypothesis is that free access to food and water limited the kidney injury, it is safe to say on page 18 lines 12-13 that suggesting dehydration and antidiuresis?

Answer: We fully agree with the reviewer for this very interesting suggestion. During the race, free access to food and water was limited to the refreshment points allowed by the race organisation. Thus, we cannot exclude that some participants experienced dehydration and antidiuresis during the race (all samples were collected before the participant refreshed). A sentence was inserted in the revised version of the manuscript to clarify this point.

Page 19 line 21 – ultra endurance races do in fact have considerable effects to kidney function such as, as the authors state, antidiuresis. However, it would be more accurate to state that ultra-endurance races specifically have no impact on acute kidney injury.

Answer: We fully agree with this remark and thank the reviewer. The sentence was modified in order to take into account the remark.

Reviewer #2: (No Response)

---

## [Editor Report · Decision Letter 2]

3 Sep 2019

Acute kidney injury during an ultra-distance race

PONE-D-19-17069R2

Dear Dr. Jouffroy,

We are pleased to inform you that your manuscript has been judged scientifically suitable for publication and will be formally accepted for publication once it complies with all outstanding technical requirements.

With kind regards,

Paolo Cravedi

Academic Editor

PLOS ONE

Additional Editor Comments (optional):

"Temporal" is an adjective. As a noun, it refers to the temples of the skull.

In the present paper, "temporal" should be used in expressions such as, for instance, "temporal changes" or "temporal variations".
---

## [Editor Report · Acceptance letter]

6 Sep 2019

PONE-D-19-17069R2 

Acute kidney injury during an ultra-distance race 

Dear Dr. Jouffroy:

I am pleased to inform you that your manuscript has been deemed suitable for publication in PLOS ONE. Congratulations! Your manuscript is now with our production department. 

With kind regards,

on behalf of

Dr Paolo Cravedi 

Academic Editor

PLOS ONE